# Sexual Dimorphism in Cardiometabolic Diseases: The Role of AMPK

**DOI:** 10.3390/ijms241511986

**Published:** 2023-07-26

**Authors:** Miroslava Kvandova, Angelika Puzserova, Peter Balis

**Affiliations:** Institute of Normal and Pathological Physiology, Centre of Experimental Medicine, Slovak Academy of Sciences, 841 04 Bratislava, Slovakia; angelika.puzserova@savba.sk (A.P.); peter.balis@savba.sk (P.B.)

**Keywords:** AMPK, sexual dimorphism, estrogen signaling, androgen signaling, endothelial function, metabolic regulation, cardiovascular diseases

## Abstract

Cardiovascular diseases (CVDs) are the leading cause of mortality and disability among both males and females. The risk of cardiovascular diseases is heightened by the presence of a risk factor cluster of metabolic syndrome, covering obesity and obesity-related cardiometabolic risk factors such as hypertension, glucose, and lipid metabolism dysregulation primarily. Sex hormones contribute to metabolic regulation and make women and men susceptible to obesity development in a different manner, which necessitates sex-specific management. Identifying crucial factors that protect the cardiovascular system is essential to enhance primary and secondary prevention of cardiovascular diseases and should be explicitly studied from the perspective of sex differences. It seems that AMP-dependent protein kinase (AMPK) may be such a factor since it has the protective role of AMPK in the cardiovascular system, has anti-diabetic properties, and is regulated by sex hormones. Those findings highlight the potential cardiometabolic benefits of AMPK, making it an essential factor to consider. Here, we review information about the cross-talk between AMPK and sex hormones as a critical point in cardiometabolic disease development and progression and a target for therapeutic intervention in human disease.

## 1. Introduction

Both men and women are affected by cardiovascular diseases (CVDs), which remain the leading cause of mortality and disability worldwide. Based on the 2020 report by the World Health Organization, around 19.1 million people, which accounts for 32% of global deaths, passed away due to cardiovascular comorbidities [1]. Indeed, the increased prevalence of overweight, obesity, and obesity–related cardiometabolic risk factors such as hypertension, glucose, and lipid metabolism dysregulation are associated with a 2-fold increase in CVD events (coronary heart disease, stroke, heart failure) risk and a 1.5-fold mortality increase [2]. Unhealthy lifestyles such as tobacco use, alcohol consumption, physical inactivity, and consuming ultra-processed foods have caused obesity rates to triple over the last 30 years [1]. Cardiometabolic diseases exhibit sexual dimorphism in their development [3]. Sex differences are apparent in the onset and prevalence of CVDs throughout life. Leening et al. [4] have shown that CVD cumulative incidence in men increased steadily with age, and in women under the age of 70, it is stable with a subsequent sharp increase [4]. Although women in the reproductive age group tend to have much lower rates of CVDs than men, these beneficial effects are not evident in patients with diabetes mellitus [5]. Due to differences in cardiovascular physiology between sexes, sex-specific management is necessary for diagnosis and treatment. Therefore, identifying critical factors that protect the cardiovascular system is crucial to improving primary and secondary prevention of cardiovascular diseases. Adenosine monophosphate-dependent protein kinase (AMPK) may be such a factor as it has been proven the protective role of AMPK in the cardiovascular system [6,7,8] and is regulated by sex hormones, both estrogen [9] and testosterone [10]. In addition, AMPK plays a crucial role in regulating numerous metabolic pathways often disrupted in diabetes mellitus. This includes triggering glucose transport in skeletal muscle and suppressing gluconeogenesis in the liver. These unique bioproperties suggest that AMPK participates in cardiometabolic protection and is a significant player. In this review, we summarized the information about AMPK as a critical link between metabolic stress and CVD development in the context of sex-specific hormone regulation.

## 2. Sexual Dimorphism in Cardiometabolic Diseases

Cardiometabolic diseases on the background of metabolic syndrome (MS), type 2 diabetes mellitus (T2DM), and hypertension exhibit sexual dimorphism in their development [3]. The development of cardiometabolic diseases is closely associated with metabolic homeostasis and its dysregulation, such as obesity, atherogenic dyslipidemia, insulin resistance (IR) with or without glucose intolerance, and pro-inflammatory and pro-thrombotic states. Significantly, more than 50% of elderly people over 65 reported obesity-related multi-comorbidities, which was even more pronounced in women. In 2018, 39% of men and 40% of women worldwide were overweight, and 11% of men and 15% of women were obese [11]. Obesity and being overweight are more prevalent in women, in most developed countries, due to fluctuating concentrations of sex hormones during women’s lives. Hormonal changes during the menstrual cycle affect caloric intake and alter 24 h energy expenditure, fat composition and distribution, adiposity, cardiac metabolism, and, subsequently, its function [12]. Similarly, the existence of sex differences in CVD occurrence is presented as well. The risk of CVDs in females is often underestimated, especially in pre-menopausal and menopausal women, due to the misinterpretation that women are still protected due to female-specific sex hormone-dependent regulation. Still, CVDs are also developed in women, but their onset is 7–10 years later than in men [13]. Although women have a lower incidence of CVDs than men, the mortality increases, and the prognosis worsens after acute cardiovascular events due to undervalued early diagnosis and less aggressive therapeutic approaches in women. It was reported that diagnostic angiograms and interventional procedures are used less frequently in women [13].

Age also plays an integral part in CVD onset. The prevalence of CVDs, such as atherosclerosis, stroke, myocardial infarction, and hypertension, is significantly higher in older adults (both men and women) than in younger populations [14]. Sex differences are apparent in the onset and prevalence of CVDs throughout life. CVD cumulative incidence in men increased steadily with age; in women, it remained low until age 70 and then increased sharply [4].

Sex differences are apparent in the onset and prevalence of CVDs throughout life; therefore, it is crucial to understand their relationship. Cardiometabolic diseases are linked to metabolic regulation, where significant sex dimorphism is noticed. Clinical results show that diabetes mellitus onset is sex-hormone-dependent [15]. Glucose homeostasis is primarily regulated by skeletal muscle due to basal and insulin-stimulated glucose uptake. The difference in muscular mass between men and women is a significant factor participating in glucose homeostasis regulation to consider. While men tend to have higher muscle mass, women display increased glucose uptake and insulin secretion due to estrogen regulation. Consequently, the distribution of adipose tissue, which varies by sex, can affect glucose metabolism. Deregulated adipose tissue expansion in favor of visceral adipose tissue is linked to IR due to an increased pro-inflammatory cytokines expression, a higher free fatty acids flow and delivery to the liver, hyperinsulinemia, and increased glucose production [16]. The expansion of adipose tissue occurs through the increase in the number of cells, either by recruiting new adipocytes (hyperplasia) or enlarging the size of cells in existing adipocytes (hypertrophy). It was reported that hyperplastic expansion results in improved metabolic health, while hypertrophic expansion can lead to systemic metabolic dysfunction [17]. It was proposed that estrogen is crucial in controlling adipose tissue expansion growth and can positively or negatively influence metabolic health [16]. Estrogen’s functions seem to have a dual role. On one side, it promotes hypertrophic-induced inflammation. On other sites, estrogen signaling initiates ubiquitination and degradation of hypoxia-inducible factor (HIF), thus inhibiting inflammation and fibrosis [18].

### 2.1. Insulin Signaling and Sex Hormones

Insulin is secreted to the blood circulation as a reaction to the postprandial increase of glycemia. Circulating insulin can bind to the insulin receptor (INSR), which all mammalian cells express. The INSR is a heterotetrameric glycoprotein consisting of two α-β dimers. Insulin binding to the α subunit leads to receptor autophosphorylation along the β subunits and subsequent insulin receptor substrates (IRS) activation. IRS represents signaling adaptor proteins linking active INSR and other components of the insulin signaling cascade. IRS1 is essential for insulin-induced glucose uptake and metabolism, while IRS2 regulates lipid metabolism [19].

Insulin resistance (IR) is a pathological condition where the normal levels of endogenous or exogenous insulin in circulation cannot regulate glucose uptake and utilization. Physiologically, insulin production and secretion from pancreatic β cells and inhibition of gluconeogenesis are initiated by increased glycemia. In IR patients, this signaling is impaired, which can cause a paradoxical increase in both hepatic glucose production and insulin secretion. This may next lead to hyperglycemia, which can be induced or worsened. Additionally, IR can impact lipid metabolism, increasing lipogenesis and a higher risk of developing nonalcoholic fatty liver disease [20]. Compromised insulin action and/or insulin secretion contribute to the setting and maintaining conditions such as hyperglycemia, hyperlipidemia, hypertension, and obesity, all of which are standard features of MS [21]. The most significant consequence of MS is an alarming increase in the risk of developing T2DM and/or CVDs [19].

The development of MS and IR affects men and women differently. Men are more likely to develop MS than women before menopause; when estrogen levels decrease, the protection in women is reduced. Several studies have shown that postmenopausal women and men in the same age group exhibit more IR than premenopausal women [22]. It has been proposed that a cross-talk exists between insulin and estrogen signaling. It seems that the crucial receptor is estrogen receptor α (ERα). It was shown that IR induced by a high-fat diet could be treated by ERα-specific agonists [23]. Similarly, ERα deficiency, whether male or female, experiences a decline in glucose regulation, resulting in higher fasting glycemia, impaired glucose tolerance, and reduced insulin-stimulated glucose uptake. Additionally, their lipid metabolism is affected, leading to hyperinsulinemia [24]. Estrogen signaling affects insulin signaling directly by regulating insulin-sensitive tissues or indirectly by regulating factors such as oxidative stress [25], contributing to IR. Estrogen signaling affects lipid metabolism as well, including cholesterol metabolism. Specifically, estrogen promotes the reverse transport of cholesterol in the liver, its conversion into bile acids, and its release into the bile duct. ERα expression in adipose tissue differs between men and women. In women, it is higher compared to men and correlates with increased insulin sensitivity. Therefore, estrogen signaling can contribute to sexual dimorphism in metabolism regulation [26].

Similarly, it was shown that androgens affect metabolism due to their involvement in lipid metabolism regulation, triacylglycerol accumulation, and adipogenesis [27]. Studies have shown that androgens possess anti-obesogenic properties. In mice that lack androgen receptors (ARs), there is a noticeable increase in adiponectin secretion, adiposity, body weight, triacylglycerols, and a decrease in insulin sensitivity [28]. Testosterone substitution therapy decreases glycemia and HbA1c concentration in men with diabetes and pre-diabetes [29]. The extent to which androgens offer protection depends on their concentration. Studies have shown that low and too-high levels of androgens are linked to metabolic dysregulation and reduced insulin sensitivity. However, it is essential to note that androgens within a specific range of normal serum levels have been shown to have protective properties for metabolism. Although it is assumed that ER mediates a significant effect on the cardiovascular system of androgens due to testosterone aromatization to estradiol (E2), it was shown that AR expressed especially in endothelial cells, cardiomyocytes, macrophages, and thrombocytes and thereby directly contributes to this protective effect [30].

### 2.2. Insulin Signaling and Metabolic “Master Switch”—AMPK

In reasoning and discussing the regulation of metabolism and the significance of insulin signaling, it is essential to acknowledge the role played by AMPK. AMPK, as a metabolic “master switch”, is a crucial element in regulating metabolism due to its high sensitivity to changes in the ratio of ATP:ADP levels and can act quickly to ensure the organism’s energy needs and survival [31]. It has been scientifically proven that AMPK displays anti-diabetic properties; specifically, it has been demonstrated that AMPK can increase the expression of glucose transporter type 4 (GLUT4) in skeletal muscle, leading to increased glucose uptake [32]. Additionally, it can promote glucose uptake due to its impact on GLUT1 [33]. Moreover, insulin sensitivity could be improved by AMPK activation due to its involvement in the regulation of expression and phosphorylation of insulin signaling members, including protein kinase-B (Akt), IRS1, IR, and acetyl-CoA carboxylase (ACC) [34].

Significantly, the action of AMPK can be regulated by both male [35,36] and female [37,38]-related sex hormones, and it was also reported that AMPK agonists could affect sex hormones metabolism and insulin signaling [39,40]. Based on the presented information, it can be inferred that a cross-talk exists between insulin signaling, sex hormones, and AMPK. The rising number of people affected by cardiometabolic diseases highlights the urgent need to explore this cross-talk and pinpoint the critical mechanisms contributing to their development and progression.

## 3. AMPK Structure

AMPK is an enzyme that Hardie et al. described in 1988 [41]. Its function was first defined as an enzyme regulating two metabolic enzymes, ACC, and 3-hydroxy-3-methylglutaryl-coenzyme A reductase (HMGR). Later, this enzyme was also studied due to its involvement in other physiological processes, such as cellular survival, especially under stress conditions [42]. Maintaining a high intracellular ATP:ADP ratio is essential for cell survival. The level of AMP increases immediately when ATP:ADP ratio decreases. The AMP increase signals that cellular energy status is at risk, and pro-survivor mechanisms should be activated where the AMPK is the central orchestrator. Once AMPK is activated, energy utilization is limited, and production is promoted to ensure cellular survival by phosphorylation of several down-stream targets [8]. AMPK has a protective role in the cardiovascular system as well. Chen et al. [43] showed, for the first time, that endothelial nitric oxide synthase (eNOS) is directly phosphorylated by AMPK at Ser1177 in endothelial cells. This finding indicates a connection with metabolic stress and the cardiovascular system [43].

AMPK belongs to the evolutionarily conserved serine/threonine heterotrimeric kinase family. This complex comprises one catalytic α subunit and two regulatory β and γ subunits. Those subunits are products of distinct genes (α1/α2; β1/β2; γ1/γ2/γ3) occurring as alternative isoforms. Twelve heterotrimeric combinations can be created by tissue-specific translational products [31]. α1, β1, and γ1 are predominantly expressed in the majority of cells. The α2 isoform is expressed in the liver [44], while the α2, β2, γ2, and γ3 isoforms are specific to skeletal and cardiac muscle [45]. Isoforms differ not only by tissue-specific expression but also by their function. The subunit combination indicates a degree of AMP dependence, mostly affected by the gamma isoform. A complex containing γ2 depends on AMP more than γ1, while a complex with γ3 is the most independent of AMP [46]. The task of the gamma subunit in the AMP dependency is characterized by the direct presence of the AMP/ATP binding domain in the domain structure [47]. Next, the combination of subunits determines the subcellular localization of AMPK. While the expression of the α2 subunit is typical for AMPK localized preferably on the nucleus, the other site, the α1 subunit, is specific for cytoplasm [31].

### 3.1. α Subunit

Alpha (α) subunit contains an N-terminal kinase domain containing Ser/Thr protein kinase, a catalytic enzyme engine (Figure 1a). This subunit includes a typical activation loop with critical Thr172 residue, which has to be phosphorylated by upstream kinases (e.g., liver kinase B1 = LKB1; Ca^2+^-calmodulin-dependent protein kinase kinase beta = CaMKKβ; transforming growth factor-β-activated kinase 1 = TAK1) to be activated. This subunit also contains a domain interacting with the C-terminal domain, which plays a structural role in the interaction of the α subunit with subunits β and γ [47]. Between those two domains is the auto-inhibitory domain (AID) formed by a three-helical bundle and α regulatory subunit-interacting motifs (αRIMs) having a decisive role in nucleotide-mediated allosteric regulation [48].

### 3.2. β Subunit

Beta (β) subunit represents a nucleus of a heterotrimeric complex because its C-terminal domain forms an extended β sheet and thus provides a scaffold for the binding of the catalytic α and regulatory γ subunits (Figure 1b). The N-terminal domain contains β sheet rich carbohydrate-binding module (CBM), acting as an allosteric inhibitor, which negatively regulates AMPK phosphorylation by upstream kinases. CBM is a part of the regulatory machinery and can capture the signal of cellular energy status through glycogen [49]. CBM structure is formed by glucose α1-6-branched glycogen and can interact with glycogen and β-cyclodextrin. This binding moves CBM away from the α kinase domain resulting in AMPK inhibition. Oppositely using pharmacological AMPK activators (991, A769662, etc.) or auto-phosphorylation of β-S108 moves CBM toward α kinase, activating AMPK [50]. N-myristoylation of β subunit N-terminal domain affects activation by upstream kinases and can mediate the interaction of AMPK with membrane and thus facilitate reversible binding of AMPK to membranes [51] or interaction with other proteins [52].

### 3.3. γ Subunit

Gamma (γ) subunit contains several domains involved in AMPK activation (Figure 1c). The C-terminal domain of all γ subunits is formed by four tandem repetitions of cystathionine β-synthase (CBS), organized into a double-stranded β sheet with two surrounding α-helices. Four CBS sites (CBS1—CBS4) form the structure so-called “Bateman” [53,54]. Every pair of CBSs create one “Bateman” structure, whose cleft possesses two adenylates (AMP, ADP, ATP) binding sites, formed by conserved Asp residues necessary for adenine ribose interaction, one on each side [54]. N-terminal domains of γ subunit are modulated by posttranslational modifications resulting in modification of protein/protein interactions and can affect subcellular localization and function of AMPK [55].

## 4. AMPK Activation

AMP is a critical activatory component. The activation of AMPK depends on AMP levels. AMP can activate AMPK by allosteric activation; moreover, its presence promotes phosphorylation of threonine residue, localized in the α subunit of the activation loop, by upstream kinases (LKB1, CaMKKβ, TAK1) [56]. AMPK activity cannot be detected without Thr172 phosphorylation [57]. Activation of AMPK due to AMP binding and/or phosphorylation processes at Thr172 leads to “active conformational changes” [50]. Furthermore, AMP promotes the inhibition of phosphatase dephosphorylating AMPK [58]. First, it was proposed that activation of AMPK is mediated because phosphorylation of AMPK by upstream kinases is simplified due to AMP binding to the Bateman domain, and such a substrate is more attractive to kinases than phosphatases [59]. Moreover, AMP binding inhibits phosphatases [60]. AMP is an ultrasensitive signaling messenger because it immediately increases when ATP:ADP ratio decreases. A slight increase of AMP levels over the critical range of concentrations can significantly increase final output—AMPK activation. A high level of ATP antagonizes the effect of AMP on AMPK. Therefore, the activation depends on the AMP:ATP ratio [61]. Interestingly, even small concentrations of AMP (50–500 μM) can lead to AMPK allosteric activation and more than 10-fold, even in the presence of much higher (more than the intracellular physiological range) concentrations of ATP (5 mM) [62].

### 4.1. Physiological Activation

AMPK can be activated physiologically by exercise or caloric restriction. Exercise represents the most extreme metabolic stress, leading to a significant increase in muscle energy turnover, alterations in nucleotide status, and thus changes in AMP:ATP ratio in skeletal muscle. AMPK is activated due to increased binding of AMP, depending on the intensity and duration of the exercise, to the γ subunit and reduced ATP binding [63]. As mentioned above, AMP can allosterically activate AMPK due to AMP binding to the Bateman domain. Still, this allosteric activation has only a small activatory potential on AMPK activation (less than 10-fold) [64]. A more significant effect of exercise-mediated AMP elevation on AMPK activation is an increase of Thr172 phosphorylation of AMPK α subunit by upstream kinases depending on AMP, where AMP increases AMPK activity more than 100-fold [65]. During basal conditions, AMPK in the muscle is continuously phosphorylated and dephosphorylated in a futile cycle, which seems energetically wasteful. Still, energy demands such as that are negligible and allow more significant and faster changes in the AMPK phosphorylation/dephosphorylation status and thus respond better and faster to various stimuli [63]. Similarly, fasting or calorie restriction can activate AMPK physiologically. Fasting creates an energy deficit due to increased ATP consumption and AMP production, resulting in changes in AMP:ATP ratio [66].

### 4.2. Pharmacological Activation

Several classes of compounds activating AMPK were described. Based on the mechanisms resulting in AMPK activation, we can divide them into five categories. **The first category** belongs to antidiabetic drugs such as metformin and berberine, inhibiting mitochondrial respiratory complex I. Administering such mitochondrial modulators increases the cellular AMP:ATP ratio and thus activates AMPK [67]. Those drugs are often used as antidiabetic drugs of the first choice due to their inhibitory effect on gluconeogenesis and insulin sensitivity improvement [68].

**The second category** of activators, e.g., 2-deoxyglucose, acts as glycolysis inhibitors, indirectly increasing the cellular AMP:ATP ratio. Increased AMP binds to γ subunit of AMPK and activates it allosterically. However, it should be kept in mind that the mechanism of action of the first and second category of activators is based on cellular ATP depletion and not on direct AMPK activation and thus are not AMPK specific [47].

**The third category** belongs to substances directly activating AMPK because they directly bind to site three on the γ subunit. Those drugs are absorbed into the cells by adenosine transporters and intracellularly metabolized by adenosine kinase to AMP analog—ZMP [68]. ZMP is a natural intermediate product of the purine nucleotide synthetic pathway. Such an activator is AICAR (5-aminoimidazole-4-carboxamide ribonucleoside) which mimics the activation process by AMP [69]. ZMP binds to AMP binding sites and triggers the corresponding functional response, such as inhibition of fructose-1,6-bisphosphatase and glycogen phosphorylase stimulation. Although ZMP imitates AMP activation, and its effectivity is 50-fold smaller than AMP, AMPK is activated by ZMP due to its intracellular accumulation (in mM concentrations) [70]. Therefore, its AMPK specificity is questioned.

A more specific AMPK activator is a small-molecule allosteric activator C13 (di-iso-propyl phosphoester). C13 represents **the fourth category** of AMPK activators. Similarly, it mimics AMP. C13 is intracellularly absorbed and metabolized by esterases to AMP analog called C2 (5-(5-hydroxyl-isoxazol-3-yl)-furan-2-phosphonic acid). C2 is a very effective activator binding to the α subunit, where isoform specificity was observed. Although it was shown that C2 is binding only to α1 and not to the α2 subunit, AMPK is also activated due to its binding to the γ subunit. Binding sites for AMP and C2 overlap but are not identical, and their allosteric effect is not additive [71]. Importantly, C2 activation of AMPK is specific and other enzymes using AMP as substrate (phosphofructokinase-1, fructose-1,6-bisphosphatase 1, glycogen phosphorylase) are not activated by C13 administration [68]. In vitro tests show that C2 is a very effective AMPK activator, where efficiency is much higher than synthetic A769662 (5th category) or AMP. Moreover, using C13 or its metabolite—C2, does not affect adenine nucleotide levels.

**The fifth category** involves synthetic compounds binding to the allosteric drug and metabolite (ADaM) site, localized between the β-CBM and the N-terminal kinase domain localized on the α subunit [48,69]. Here belong compounds such as A769662 (thienopyridine), 991 (benzimidazole), MT63-78, PF-249, etc. They activate more AMPK-β1 rather than AMPK-β2. The degree of AMPK activation depends on Thr172 phosphorylation, which is essential for AMP-dependent AMPK activation. In the case of Thr172 phosphorylation, activation of AMPK due to the binding to the ADaM site results in modest allosteric activation going along with an inhibitory effect on dephosphorylation and promoting Thr172 phosphorylation [72]. More significant allosteric activation (65-fold) by the fifth category of AMPK activators is observed when Thr172 is not phosphorylated, and this effect is synergic with AMP [73]. Moreover, activation by A769662 and 911 requires Ser108 phosphorylation of the β subunit. Based on the fact that the synergic effect of AMP and A769662 was observed, the binding sites of AMP and A769662 are not the same, and activatory mechanisms are also different. Compared to AICAR, activation by A769662 is very specific to AMPK, and it does not activate other enzymes using AMP as a substrate [68].

### 4.3. Activation by Upstream Kinases

To promote AMPK activity due to nucleotide binding to the γ subunit, AMPK must be phosphorylated on the Thr172 phosphorylation of AMPK α subunit, and without this “activatory action”, no detectable activity of AMPK can be described [57]. In cells with deletion of kinases most responsible for AMPK activation, Thr172 phosphorylation is almost wholly abolished, and AMPK activity is undetectable. Even though some AMPK activators (AMP, A769662, 991) do not require AMPK phosphorylation, the crystal structure of non-Thr172 phosphorylated but activated complex of AMPK—AMP/991 is similar to the activated Thr172 phosphorylated complex. The binding of such activators initiates conformational changes in the activator loop, and therefore AMPK activation without Thr172 phosphorylation can be observed. However, it should be mentioned that this type of AMPK activation is significantly less efficient than phosphorylated AMPK, and phosphorylation is necessary for intracellular activation of AMPK signaling [74]. Phosphorylation by upstream kinases represents an essential interplay between stress conditions (often stress-induced stimuli activate upstream kinases, e.g., increased intracellular Ca^2+^) and activation of AMPK. This mechanism puts AMPK in the position to receive information from several cellular signaling pathways and act to secure cellular survival [61].

Several upstream kinases phosphorylating AMPK were described, such as LKB1 [75], calcium/calmodulin-dependent protein kinase kinase 2 (CaMKK2, CaMKKβ) [61], and TAK1 [76].

As is known, CaMKK2 belongs to the multi-functional Ser/Thr protein kinases family. The function of CaMKK2 depends on intracellular calcium levels. The role of upstream kinases is tissue-specific, and CaMKK2 activates AMPK by α subunit phosphorylation primarily in particular cell types, including endothelial cells, hypothalamic neurons, and T-cells and based on this tissue-specific action is CaMKK2 also tissue-specific expressed [77]. The cellular microenvironment is a limiting factor activating CaMKK2. Here, calcium influx is a vital signaling messenger. It was shown that calcium dysregulation is present during cardiometabolic disorders [78].

LKB1 is Ser/Thr protein kinase, which by the function associated with energy metabolism regulation due to AMPK activation, and it is proposed that LKB1 is the master upstream kinase phosphorylating AMPK [8,79]. LKB1 is permanently active and ubiquitously expressed kinase. Under basal conditions is localized in the nucleus, and its catalytic function is significantly limited. Its activity depends on translocation to the cytosol, where it creates a heterotrimeric complex with STRAD (sterile-20-related adaptor) and MO25 (mouse protein-25) [80]. Such a constitutive active heterotrimeric kinase activates AMPK by Thr172 phosphorylation [80,81]. Since LKB1 is ubiquitously expressed and permanently active, the activation of LKB1 depends on the presence of its heterotrimeric partners [80] and post-translational modifications (ubiquitination, deacetylation, oxidative modification, etc.) [81].

Another upstream kinase phosphorylating AMPK is TAK1. Although its role in AMPK activation remains unclear, therefore opinions of many scientific teams on the issue differ [76]. TAK1 was identified as a serine/threonine protein kinase family and a MAPK kinase (MAP3K) family member. TAK1 is ubiquitously expressed during early development, and at mid-gestation, its expression becomes more restricted. Increased expression of TAK1 was observed in a tissue-specific manner during the development of organs, such as kidneys, liver, testes, nervous system, etc. TAK1 is activated by transforming growth factor-β, cytokine regulating a wide range of intracellular signaling pathways. Moreover, it can be activated by other (patho-) physiological stimuli such as lipopolysaccharides, pro-inflammatory cytokines (interleukin 1), tumor necrosis factor α (TNFα), and others [82]. TAK1 was identified as AMPK activating upstream kinase. Although several authors described its activatory role, the direct activatory mechanism was questioned because, in some studies, the involvement of LKB1 as an intermediate kinase was observed [83]. Despite these contradictions, it is undeniable that TAK1 is an important AMPK regulator regardless of whether AMPK is activated directly or indirectly via the intermediated LKB1 [76].

### 4.4. Activation by Intracellular Physiological Stimuli

The AMP-dependent protein kinase could be activated by several physiological stimuli associated with cellular stress, such as hypoxia, shear stress, low glucose, oxidative stress, etc. Similarly, some other cardiovascular stimuli were reported to activate AMPK. It was shown that AMPK is activated after administering thrombin, vascular endothelial growth factor (VEGF), bradykinin, and estrogen. Here, AMPK activation is mediated due to Thr172 phosphorylation by upstream kinases [8].

Several studies pointed to the cross-talk of AMPK and reactive oxygen species (ROS) production [6,84]. Opinions on AMPK activation and allosteric activation by AMP due to increased ROS differ. Some authors showed that ROS could independently on AMP:ATP level changes activate AMPK, particularly H_2_O_2_, due to the Thr172 phosphorylation by upstream kinases [85]. Moreover, it was observed that H_2_O_2_ administration causes oxidative modification of cysteine residues (Cys299 and Cys304; for AMPK activation critical cysteine residue) of AMPK α subunit and thus activate AMPK independently on AMP:ATP changes [86]. Other authors dispute these results and claim that the main driving forces of AMPK activation are really changes in adenine nucleotides, mainly due to changes in mitochondrial ATP production [87].

Another physiological stimulus activating AMPK is hypoxia. Hypoxia is a cellular state where an insufficient amount of oxygen is present. Complete oxygen depletion leads to irreversible cellular damage, and cell death is initiated. The mechanism responsible for cellular adaptation and, thus, ensuring cell survival is HIF-1 [88]. Here, several possible mechanisms of AMPK activation were described, and they depend on the extent of hypoxia. First, decreased ATP production due to hypoxia-induced β-oxidation inhibition was observed, leading to changes in AMP:ATP ratio and allosteric AMPK activation [89]. Second, during hypoxia, ROS are produced in an extended manner, mainly by the mitochondrial electron transport chain [90]. ROS is a link to several physiological stimuli associated with stress conditions (metabolic stress, inflammation, hypoxia, etc.) and AMPK activation. As previously stated, AMPK activation is triggered by oxidative stress through various means, including direct oxidative modification of cysteine residues, direct allosteric activation, and phosphorylation of Thr172 [8].

### 4.5. AMPK as a Mediator of Hormonal Signaling

It has to be mentioned that as a master energy homeostasis regulator, AMPK is the mediator of several hormonal signals such as leptin, adiponectin, insulin, etc. AMPK is expressed in nervous tissue and affects neuroendocrine function, thus controlling food intake [91]. Hypothalamic AMPK activity is changing along with changes in cellular nutrient and glucose levels. It increases during fasting and decreases under conditions of *ad libitum* saturation [92]. Moreover, hormones that the gastrointestinal system produces (peptide YY, ghrelin, glucagon-like peptide 1, etc.) and adipose tissue (leptin, resistin, adiponectin) regulate AMPK activity [91]. The AMPK activity can be controlled by sex hormones as well.

#### 4.5.1. Female-Related Hormones: Estrogen

Estrogens are synthesized mainly by the gonads, specifically the ovaries and testes. They also produce extragonadally, including the adrenal glands, brain, adipose tissue, skin, and pancreas [93]. Undoubtedly, 17β-estradiol (E2) is the primary sex and a steroid hormone in females. Estrogens (also including estrone—E1 and estriol—E3) are essential in regulating various physiological functions, particularly those related to reproduction in both men and women. Specifically, this refers to the development and function of reproductive organs and the emergence of secondary sexual characteristics. In addition, estrogen’s non-reproductive functions were described, particularly in the physiology of the cardiovascular, skeletal, muscular, immune, endocrine, and central nervous systems [94]. Notably, estrogen also plays a role in energy metabolism. Low estrogen levels are associated with weight gain and subsequent obesity due to increased food intake and decreased energy expenditure [95]. Regulating metabolic functions via E2 is crucial, especially regarding sexual dimorphisms, such as body weight, food intake, glucose, and lipid homeostasis. Lowering circulating E2 can result in significant metabolic changes, leading to altered fat distribution, weight gain, inflammation, abnormal lipid profile, and diminished insulin sensitivity [96]. Research on animal models of obesity has revealed that taking E2 supplements could help alleviate metabolic abnormalities [97].

Several receptors mediate the physiological action of E2. The genomic effect of estrogen is mediated through nuclear estrogen receptor (ERα, ERβ) activation. Once these receptors have homo- or heterodimerized, they proceed toward the nucleus to attach themselves to estrogen response elements (ERE). Although ERα and ERβ share a similar structure, they exhibit distinct DNA and ligand binding domains, resulting in different transcriptional effects. The non-genomic effect of estrogens is triggered by the extracellular estrogen receptor and an orphan—G-protein-coupled estrogen receptor (GPER, previously known as GPR30) [98]. It was observed that E2 could activate several signaling cascades as mitogen-activated protein kinases ERK1/2 and -p38 or phosphoinositide 3-kinase-serin/threonine-specific kinase B (PI3K/AKT) [99]. Multiple estrogen receptors allow for the regulation of metabolism through both genomic and non-genomic pathways. These receptors can act synergistically or antagonistically, depending on the conditions. ERα or *GPER* deletion in animal models results in similar metabolic phenotypes, indicating that both receptors may collaborate to regulate metabolic effects through shared or distinct pathways [100]. Although the role of ER in metabolism regulation is established, the part of GPER remains unclear and requires further clarification.

It was shown that estrogens can activate AMPK (Figure 2). Here, various mechanisms have been proposed. Yang and Wang [37] showed that estrogen activates AMPK by Ca^2+^-induced CaMKK2-dependent phosphorylation in endothelial cells [37]. E2-activated ERs (ERα, ERβ, and GPER) regulate intracellular calcium concentrations along with β-adrenergic receptors [101]. Estrogen’s regulation of ion channels is mediated through binding membrane receptors, such as G-protein coupled receptors or insulin-like growth factor receptors. This leads to subsequent activation of second messenger and/or ion channels targeting signaling pathways [102]. The phosphorylation of L-type Ca^2+^ channels is followed by the entry of intracellular Ca^2+^ and the release of Ca^2+^ by the sarcoplasmic reticulum [103]. Activated CaMKK2 can then phosphorylate AMPK as described.

However, AMPK can also be activated by Sirtuin 1 (Sirt1; also known as NAD-dependent deacetylase sirtuin-1)-dependent LKB1 deacetylation. Sirt1 is a Type III histone deacetylase that, when overexpressed, can decrease the Lys48 acetylation of LKB1. This, in turn, increases the activity of LKB1 [38]. It was shown that E1 and E2, but not E3, increase expression of Sirt1 via active ERs cellular translocation and binding to ERE located in the Sirt1 promotor region [104].

It appears that estrogen plays a significant role in influencing the differences between males and females in terms of their cardiovascular systems [105]. It is known that menopausal women, due to a significant reduction of estrogen production by ovarian follicles, are at higher risk of CVDs development than men of the same age [106]. Estrogens protect against CVD through their role in vascular and cardiac physiology. Acute E2 administration due to the eNOS activation rapidly increases coronary blood flow. Here, ERα activation induces its association with the p85α subunit of phosphoinositol 3-kinase (PI3K) and the subsequent activation of protein kinase B (AKT). The activation of eNOS at Ser1177 through phosphorylation is primarily facilitated by AKT phosphorylation. AKT also affects eNOS sensitivity to calcium [7]. Moreover, estrogen protects arteries against hypertension-induced vascular damage [107]. It was shown that estrogen also has an anti-atherogenic effect and affects inflammation by preventing abnormal adhesion of monocytes and transmigration in the vasculature [108].

#### 4.5.2. Male-Related Hormones: Androgens

Testosterone is the primary male sex hormone, along with its bioactive metabolites, such as dihydrotestosterone (DHT). DHT is the most active androgen synthesized from testosterone by 5α-reductase. Both men and women produce androgens, which are 19-carbon (C19) steroid hormones produced by the adrenal glands and gonads from C21 precursor steroids. They can also be converted into C18 steroids, known as estrogens [109]. Androgens play a significant role in developing both primary and secondary sexual characteristics. Additionally, research has shown that they regulate anabolic functions, including increased muscle mass and bone density. Interestingly, androgen deficiency due to primary or secondary hypogonadism, physiologically due to aging (men over 40), or pathophysiologically due to obesity, are associated with MS, T2DM, and IR [110]. The metabolic control by androgens is mediated by the regulation of glucose and lipid homeostasis, insulin and leptin sensitivity improvements, and regulation of pancreatic β–cell function to improve glucose tolerance and glucose-stimulated insulin secretion. Moreover, androgens prevent visceral fat accumulation and increase energy expenditure [109].

Precursors of androgens (dehydroepiandrosterone and androstenedione) are secreted chiefly by adrenal glands. In women, testosterone is secreted by ovarian theca cells; in men, it is excreted in testicular Leydig cells. In addition, peripheral tissue and adrenal glands can produce testosterone [110]. Subsequently, it is transported via the carrier protein, sex-hormone binding globulin (SHBG). Only a small percentage of testosterone (1–2%) is unbound and is the most biologically active [111].

A family of androgen receptors (ARs) mediates the action mechanism of androgens. Furthermore, it was shown that androgens could also act by activation of ER due to testosterone aromatization to E2 [111]. Testosterone and its metabolically more active metabolite, dihydrotestosterone, bind to cytoplasmic AR, chaperoned by heat shock proteins. ARs are expressed in many tissues with the highest concentrations in the male reproductive system [112]. Both genomic and non-genomic actions of AR were described. Once the active complex is translocated to the nucleus, forming a dimeric complex with another androgen-AR complex and coactivator proteins resulting in the transcription of genes, this genomic effect is relatively slow, and minutes to hours are necessary to achieve full impact [113]. Additionally, it has been observed that AR exhibits non-genomic effects, especially in excitable tissues such as the central nervous system and reproductive and non-reproductive smooth muscle [109]. In particular, androgens have a direct and immediate effect (within seconds) on membrane-binding proteins that are capable of binding testosterone, such as the ZIP9 Zinc transporter and SHBG, and other membrane-binding proteins, such as G-protein-coupled receptors, leading to the activation of membrane calcium channels and activation of secondary messengers such as inositol trisphosphate and diacylglycerol. This can ultimately alter cytoplasmic calcium and potassium channel activity [109,111].

A down-stream pathway of AR was identified as the signaling of CaMKK2—AMPK (Figure 2). AR activation can lead to calcium channel phosphorylation, and the importance of Ca^2+^ in activating AMPK is suggested that androgens can activate AMPK via CAMKK2 [35]. In addition, research has shown that testosterone can activate AMPK through LKB1 kinase. Mitsuhashi et al. [36] demonstrated that administering testosterone led to an increase in LKB1 phosphorylation, which in turn activated AMPK. However, the effect was weakened when siRNA for LKB1 was utilized [36]. The mechanisms mentioned are a result of the non-genomic impacts of activating AR. It has not yet been described how the genomic effects of AR may activate AMPK. However, further studies are necessary to establish the connection between androgens and AMPK signaling.

Notably, an association was noted between the development of CVDs and testosterone levels. Studies have shown a correlation between testosterone deficiency and metabolic imbalance. Patients with this deficiency may experience visceral obesity, IR, and an increased risk of T2DM [114]. On the other hand, excessive activation of ARs can result in an imbalance in metabolic homeostasis. The protective effect of testosterone is concentration dependent, with the most effective results being achieved within a specific range of normal serum testosterone levels. High-dose testosterone administration impaired insulin sensitivity in castrated males. In contrast, a low testosterone concentration improved insulin sensitivity [115]. Similarly, the elevated testosterone level in women increases their risk of developing T2DM. Furthermore, several studies indicate a positive correlation between testosterone levels in women, IR, and impaired glucose tolerance. For women with high levels of androgens, also known as hyperandrogenemia, antiandrogenic treatment can help partially prevent IR. Those facts point to the causal role of hyperandrogenemia and IR [116]. Likewise, there was also an observed correlation between CVD risk and testosterone level. Several authors showed that the average endogenous testosterone level has a cardioprotective effect and can be used as a therapeutic tool. In addition, in CVD patients, a reduced testosterone level was observed [117]. Reduced testosterone was associated with an increased incidence of atherosclerotic plaque, endothelial dysfunction, and inflammation [118]. Additionally, several studies have demonstrated that reduced testosterone is a risk factor for developing various diseases, such as T2DM, MS, hypertension, stroke, dyslipidemia, renal failure, malignancy, etc. Additionally, it is essential to note that patients with low testosterone levels had significantly higher mortality rates for the mentioned diseases [111].

### 4.6. The Regulation of Sex Hormones by AMPK

As previously mentioned, sex hormones have the ability to activate AMPK and thus regulate metabolic homeostasis. However, research has also demonstrated that AMPK can regulate sex hormones activity (Figure 2). Although AMPK activators, such as metformin and resveratrol, reduce the production of estrogens [39], a study by Sopariwala et al. [119] proved that AMPK signaling can regulate the expression and activation of ERs. The activation of AMPK by ZMP analog, AICAR, showed a significant increase in ERα levels in skeletal muscle. Conversely, using an AMPK inhibitor, Compound C resulted in a substantial down-regulation of ERα. Moreover, the down-regulation of ERα was observed in the mice lacking both *AMPK α1*/*α2* subunits [119]. However, it still is not clear how those activators regulate the expression and activation of ERs. On the contrary, it is known that the AMPK activator, resveratrol, shares a structural resemblance with estrogens and is categorized as a natural product belonging to the group known as phytoestrogens and can activate ERs directly by binding to nuclear ERs, membrane ERs, or extracellular GPER [120].

Similarly, AMPK activators were observed to reduce the serum testosterone level in vivo and in vitro [39,40]. Here, the action of metformin is mediated by the modulation of androgen biosynthesis. In particular, metformin inhibits the activity of 17α-Hydroxylase/17,20 lyase and 3β-hydroxysteroid dehydrogenase type 2. Interestingly, this metformin action involves inhibiting mitochondrial complex I, not AMPK signaling [40]. In addition, research has demonstrated that metformin can lower the expression of ARs through disruption of the AR and proteins midline-1 complex, resulting in the release of associated *AR* mRNA and consequent down-regulation of AR protein. This effect is not dependent on AMPK [121]. Based on the observations, it seems that AMPK signaling is affected by sex hormones and can also control estrogen and androgen signaling. However, the underlying mechanisms can vary, whether AMPK-dependent or independent, depending upon the activator employed.

## 5. Sexual Dimorphism in AMPK Regulation of Endothelial Function

### 5.1. Endothelial Function

At present, it is well-recognized that endothelial dysfunction is a crucial player in the CVD pathogenesis [122]. Maintaining vascular homeostasis is crucial for cardiovascular health [123,124], and its disturbance is a significant risk factor for CVD-related morbidity and mortality [122,125,126,127]. It proposed that endothelial dysfunction links CVD event risk and metabolic disturbance. The combination of metabolic and hemodynamic abnormalities and inflammatory processes synergically threaten the vascular endothelium, thus leading to endothelial dysfunction [128]. The proper functioning of vascular endothelium primarily depends on the bioavailability of NO, which is determined by the equilibrium between its production in the blood vessels and its degradation by ROS. ROS are physiologically produced by aerobic metabolism. An antioxidant defense system usually regulates its concentration to maintain the balance between the physiologically necessary ROS values and the production of ROS, where they already have adverse effects leading to cell damage and death. Several producers generate ROS. In the context of vascular homeostasis, are ROS mainly produced by nicotinamide adenine dinucleotide phosphate oxidases (NOX) at the plasma membrane, by the mitochondrial electron transport chain (ETC), specifically through complexes I and III, through univalent reduction of molecular oxygen caused by leaked electrons, and uncoupled eNOS [129,130]. ROS generation by mitochondrial complex I through reverse electron transport is particularly noteworthy as a potential mitochondrial redox signal because its magnitude responds sensitively to redox status and occurs under physiological conditions [86]. The mitochondria are the primary source of intracellular ROS levels and contribute up to 95% of total ROS levels [131]. It is observed that mitochondrial dysfunction and mitochondrial ROS production is increased due to metabolic misbalance during cardiometabolic diseases such as T2DM and MS [132]. Several studies pointed out that metabolic imbalances and impaired insulin signaling can cause the development of endothelial dysfunction. This is due to the alteration of serine/threonine kinases by ROS. When IRS protein undergoes increased serine phosphorylation, it results in decreased activity of insulin down-stream signaling pathways such as PI3K and AKT. This, in turn, leads to reduced eNOS activation and NO production and/or eNOS uncoupling, increased vascular smooth muscle calcium sensitization, and reduced vasodilation [132].

### 5.2. AMPK and Endothelial Function

The regulation of vascular function depends mostly on endothelium-derived NO, as it plays a crucial molecular role in this process. Not only does it help to relax vessels, but it also has beneficial anti-inflammatory and antithrombotic effects in the vasculature. A decrease in the production of endothelium-derived NO can result in endothelial dysfunction, commonly observed in conditions such as hypertension, diabetes, and atherosclerosis [133]. Various factors can impact the activity of eNOS, such as post-translational modifications, protein–protein interactions, cofactors and prosthetic groups, calcium/calmodulin, and phosphorylation [134]. AMPK plays a role in the regulation of these processes. AMPK can be activated by several stimuli such as shear stress, statins, exercise, fasting, and hormonal stimuli and can phosphorylate eNOS at Ser633/635. This, in turn, enhances the availability of NO and/or the antioxidant potential of endothelial cells [135,136,137]. Moreover, AMPK modulates pathways responsible for NOS cofactor availability. The proper functioning of eNOS depends on the levels of intracellular tetrahydrobiopterin (BH4), which is crucial for dimerization. BH4 levels are dictated by a balance of de novo synthesis regulated by GTP cyclohydrolase I (GCH I), BH4 oxidation, and recycling of BH2 to BH4. AMPK suppresses the 26S proteasome-dependent GTPCH I degradation in vitro, reversing diabetes-induced endothelial dysfunction [138].

AMPK also regulates endothelial function due to its role in endothelium-dependent hyperpolarization (EDH). It was shown that impaired EDH-type relaxations, along with increased arterial blood pressure, are present in endothelial-specific *αAMPK* knockout mice [139]. Impaired EDH relaxation is associated with the reduced engagement of small-conductance Ca^2+^-activated potassium channels in favor of intermediate-conductance Ca^2+^-activated potassium channels, which actions depend on AMPK and Sirt1 [140]. By understanding the functions of AMPK and Sirt1 in EDH-like reactions, effective pharmacological strategies can be identified to prevent vascular complications resulting from various causes. Moreover, it has already been demonstrated that estrogens can induce the activation of Sirt1/AMPK signaling in endothelial cells [104]; therefore, the clarification of these mechanisms could provide valuable insights into the role of sex differences in the development of endothelial dysfunction and inform the use of sex-specific pharmacological therapies.

Next, it has been reported that endothelial dysfunction is associated with high uric acid (UA) levels. The mechanism by which hyperuricemia induced-endothelial dysfunction contributes to CVDs is not yet fully understood. Specifically, elevated UA levels have been found to inhibit eNOS activity and decrease NO bioavailability [141,142]. Moreover, UA-induced endothelial dysfunction was associated with decreased intracellular ATP levels in endothelial cells [143]. Notably, AMPK activity is reduced in fatty liver, and UA inhibits AMPK activity in fructose-fed hepatocytes [144]. It was also shown that metformin protects against IR induced by high UA in cardiomyocytes, improves insulin tolerance and glucose tolerance in an acute hyperuricaemic mouse model, along with the activation of AMPK [145]. According to sex differences, women showed lower levels of UA than men [146]. However, there needs to be more extensive research on the sex-specific association between AMPK, UA, and endothelial dysfunction [5].

#### 5.2.1. AMPK and Antioxidant Defense as a Protective Tool for Endothelial Homeostasis Maintenance

The bioavailability of eNOS is modulated by antioxidant and anti-inflammatory actions regulated by the AMPK. AMPK activation improves mitochondrial function by its down-stream target—Peroxisome proliferator-activated receptor-gamma coactivator 1 alpha (PGC1α) and subsequent initiation of antioxidant defense (e.g., superoxide dismutase 2 = SOD2) and anti-inflammatory response [147]. Schulz et al. [148] showed that AMPK activation by AICAR (200 mg/kg/day) led to a significant improvement in endothelial dysfunction, diminished superoxide production in the vascular wall, and a reduction of myocardial NADPH oxidase activation caused by angiotensin II (AT II) infusion (1 mg/kg/day) [148]. The explicit role of AMPK in vascular homeostasis was also proved by animal studies using global *AMPK* knockout. In an animal model with a global deletion of *α1AMPK*, even “low doses” of AT II (0.1 mg/kg/day) led to the development of endothelial dysfunction, increased oxidative stress, and inflammatory response due to increased *NOX2* mRNA and protein expression [6].

AMPK is an enzyme sensitive to redox homeostasis changes and can act as a powerful trigger for initiating an antioxidant response. In the last ten years, multiple research studies have concentrated on the antioxidant properties of AMPK. It was reported that AMPK activity could be regulated directly by oxidation of critical cysteine residues, e.g., peroxiredoxin dimerization [85], and/or indirectly by redox signaling/regulation [149,150].

The master regulator of antioxidant response is nuclear factor erythroid 2-related factor 2 (Nrf2). Nrf2 belongs to the cap “n” collar family of transcription factors that contain leucine zipper regions required for antioxidant response element (ARE) binding. They are essential in regulating cytoprotective enzymes, including the antioxidant and anti-inflammatory defense machinery, maintaining cellular redox balance. Nrf2 is activated as part of the cellular response to oxidative stress induced by various physiological or pathological circumstances [151]. Its activation is triggered by ROS-dependent phosphorylation of protein kinase Cδ (PKCδ) at Ser40, resulting in the release of Nrf2 from the cytoplasmic Keap-1/Nrf2 complex, and Nrf2 is then as active transcriptional factor translocated into the nucleus [152]. Moreover, the Keap-1/Nrf2 complex is disrupted, and active Nrf2 is released due to the direct oxidation of Keap-1 thiol groups. Furthermore, the activation of Nrf2 is regulated by glycogen synthase kinase 3β (GSK3β), which catalyzes the inhibitory phosphorylation of Nrf2 by nuclear exclusion [153]. AMPK activates Nrf2 by direct Ser550 phosphorylation. In addition, AMPK inhibits GSK3β [154]. The interplay between redox and metabolic signaling pathways has become increasingly important, and communication between critical regulators AMPK and Nrf2 seems crucial. This direct redox-metabolism connection is essential for preventing cardiometabolic disease, as AMPK is a vital regulator of oxidative stress and defense [154].

#### 5.2.2. AMPK and Anti-Inflammatory Actions as a Protective Tool for Endothelial Homeostasis Maintenance

Increasing evidence suggests that understanding the metabolic pathways in immune cells is essential for a comprehensive understanding of the pro- and anti-inflammatory response. Recent studies have revealed that AMPK plays a crucial role in bridging the gap between metabolism and inflammation by controlling the fate and function of immune cells. It has been confirmed that AMPK can control inflammatory processes to protect vascular homeostasis. Several authors have reported the multi-functionality of AMPK in regulating endothelial homeostasis with emphasis on the anti-inflammatory properties of AMPK [8,155,156]. The α1AMPK, particularly in endothelial cells, appears vital in attracting inflammatory cells to the blood vessels, regulating inflammatory reactions and vascular function during oxidative stress conditions. Loss of α1AMPK impairs endothelial cell barrier function, leading to increased recruitment of inflammatory cells and upregulation of cytokines and inflammatory proteins (MCP-1, chemokine C-C chemokine receptor type 2 = CCR2), which was accompanied by an augmented production of vascular ROS generated by NOX2. Moreover, the oxidative stress induced by AT II was supported by the down-regulation of antioxidant and cytoprotective enzyme, heme oxygenase 1 (HO1), in *α1AMPK* deficient cells [156].

Among immune cells, AMPK is essential for T-lymphocyte function, with the α1 subunit being crucial for the CD8 T cell proliferation and differentiation. AMPK links nutrient availability to T-cell effector function, and α1AMPK is essential in metabolic adaptation for T-cell response during inflammation. Moreover, it described the role of α1AMPK in myeloid cells. The role of α1AMPK is crucial for proper macrophage function, representing an essential factor for macrophage polarization. In mice with macrophage-specific *α1AMPK* deletion, macrophages are kept in the M1 state, and the amount of M2 macrophages is reduced. That affects the cytokine profile produced by macrophages in favor of pro-inflammatory cytokines and induction of the pro-inflammatory phenotype, and phagocytosis of necrotic and apoptotic cells [8]. However, when AMPK is activated, it effectively suppresses the inflammatory response and leads to a notable decrease in the release of pro-inflammatory IFN-γ and TNF-α [157]. Based on that, it is undeniable that AMPK is involved in atherosclerosis. AMPK activation affects macrophage-derived foam-cell function by reduction of LOX-1 expression and oxLDL uptake in macrophages due to p300 HAT acetylation and thus suppression of NF-kB signaling pathway and inhibition of monocyte adhesion and VCAM-1 expression [158]. Even though there are some encouraging outcomes, we are still unsure about the effects of AMPK activity on immune cell function. Further research is required to shed light on this critical topic.

#### 5.2.3. Endothelial Function, Sex Hormones, and Cross-Talk to AMPK

Moreover, it was proven that sex hormones could affect endothelial function in a wide range. The regulation of endothelial function through hormones can occur via receptor-dependent or receptor-independent mechanisms. Human vascular endothelium, smooth muscle cells, macrophages, and platelets have all been found to express ERs and ARs [159]. Genomically, endogenous estrogen positively impacts vascular endothelium, including NO production modulation due to eNOS expression and/or phosphorylation. This results in endothelial vasodilatation, protection against lipid oxidation, reduced oxidative stress, and protection against vascular inflammation. Estrogen also favors cardiometabolic risk factors, including diabetes, hypertension, and MS, contributing to vascular endothelium damage [160]. In addition, estrogen signaling has a non-genomic effect on regulating endothelial function through interactions with membrane receptors or protein receptors associated with steroid hormones. This process involves the activation of common second messengers, such as intracellular calcium and cyclic AMP [159]. In considering the physiological effects of androgens, it is crucial to consider the specific tissue distribution of enzymes responsible for metabolizing testosterone, as aromatase metabolizes testosterone to estrogen, and thus androgens can also activate ERs. Interestingly, the localization of aromatase in blood vessels indicates that the vascular effects of circulating testosterone may involve both androgenic and estrogenic pathways. In male blood vessels, estrogen produced by aromatase can have a significant impact on endothelial function and the production of NO. Studies in male aromatase knockout mice suggest that estrogen produced from testosterone influences the activity of NOS. In aromatase knockout mice, was observed impaired NOS-dependent endothelial relaxation. Thus, testosterone may affect endothelial NO production indirectly through aromatase-dependent metabolism to estrogen and subsequent stimulation of eNOS signaling [161]. Some studies have shown a direct effect of testosterone on NO signaling via the ARs [162]. However, due to the possible conversion of testosterone to estradiol, these studies do not definitively confirm that androgens directly modulate endothelial function by activating the AR, and further research is needed.

Considering the AMPK, it modulates all mentioned pathways affecting vascular endothelium. AMPK activates the protective PI3K/AKT/eNOS signaling pathway [163]. Additionally, AMPK is crucial in eNOS activation by promoting eNOS and HSP90 interaction necessary for maximal eNOS activation [7]. Furthermore, AMPK positively affects angiogenesis [164]. According to an animal model study that lacked *α1AMPK*, even small amounts of AT II (0.1 mg/kg/day) caused endothelial dysfunction, elevated oxidative stress, and induced vascular inflammation in the aorta. Meanwhile, the activation of AMPK through AICAR treatment results in the restoration of vascular function, which AT II previously disrupted, and the decrease of endothelial cell apoptosis [6]. On the other hand, control of oxidative stress by AMPK activation restored normal estrogen responses in cultured human endothelial cells, even in the presence of hyperglycemia [5]. All those facts suggest that the cross-talk between sex hormones and AMPK could be responsible for cardiovascular protection (Figure 3).

## 6. Sexual Dimorphism in AMPK Regulation of Cardiac Function

The beneficial role of AMPK is not only present in endothelial cells but also in cardiomyocytes. The metabolic demands of the heart depend on the cardiac workload, hormonal status, etc. Cardiomyocytes are metabolically versatile cells that can use various fuel sources to generate ATP, including fatty acid oxidation (50–70% of their energy production), glucose oxidation, the tricarboxylic acid cycle, and amino acids [165]. AMPK plays a crucial role in regulating the energy balance of cardiomyocytes. It acts as a metabolic sensor, coordinating the anabolic and catabolic processes within the cell by phosphorylating different proteins involved in metabolic pathways and mitochondrial biogenesis in cardiomyocytes [166]. Although the focus has been on the role of AMPK in regulating cardiac energy metabolism [167,168], this enzyme also plays a crucial part in various other cellular processes that are not directly related to ATP production, e.g., post-translational acetylation, autophagy, mitochondrial autophagy, endoplasmic reticulum stress, and apoptosis [166,169].

The activation of cardiac AMPK is regulated by LKB1 phosphorylation. LKB1 over-expression promotes cardiac AMPK activation, while cardiomyocyte-specific deletion of LKB1 results in impaired AMPK activation and, thus, cardiac dysfunction [170]. Animal experimental studies have shown that cardiomyocyte-specific *LKB1* deletion, an upstream kinase for AMPK, can cause hypertrophic cardiomyopathy and left ventricular contractile dysfunction. Those animals also exhibit an elevated risk of atrial fibrillation [171]. Metformin, empagliflozin, and resveratrol treatment have improved cardiac function due to AMPK-mediated cellular metabolism regulation, antioxidant response initiation, and hypertrophy suppression. Additionally, these treatments have been observed to enhance myocardial efficiency by modulating the function of contractile proteins and regulating the expression of electrical channels [171]. All those facts especially emphasize the protective role of AMPK in the cardiovascular system.

Cardiac pathology is thought to be significantly impacted by mitochondrial dysfunction. Cardiomyocytes have a high energy demand, which is reflected in the fact that approximately one-third of their cell volume is occupied by mitochondria. These mitochondria produce over 95% of the ATP in the myocardium. They also help regulate redox status, calcium homeostasis, and lipid synthesis [172]. The mitochondria are the primary source of ATP in cells. AMPK acts as a sensor for ATP levels and, therefore, can be activated by factors such as inhibitors of respiratory chain complexes, proton ionophores, and depletion or mutation of mitochondrial DNA [173].

There is a strong indication that sex hormone signaling plays a role in mitochondrial dynamics and cellular redox biology, as mitochondrial physiology and gene expression related to mitochondria show significant differences between males and females [174]. Sex hormone receptors alter mitochondrial dynamics by modulation of gene expression [175]. Studies have demonstrated that androgen signaling affects mitochondria in some cell types, such as sperm cells, but not in cardiomyocytes [174]. On the other hand, the impact of estrogen on mitochondria in cardiomyocytes has been extensively researched [176]. Research on animals has revealed that estrogen and estrogen-related receptors are essential to maintain the proper functioning of the heart by regulating mitochondrial processes [177]. Research has indicated that females possess lower mitochondrial content but higher mitochondrial efficiency and differentiation, along with higher glutamate/malate-stimulated respiration, higher ADP/oxygen ratios, lower ROS production, lower calcium uptake, and higher calcium retention. These differences between sexes indicate a potential role for ER signaling in mitochondria [174,178]. Mitochondria play a crucial role in connecting AMPK signaling, sex hormones, and cardiovascular protection, considering their sex-specific regulation of mitochondrial homeostasis and AMPK-dependent regulation of mitochondrial metabolism.

## 7. Conclusions

Over the last decades, there has been a heightened focus on understanding cardiometabolic pathways to protect vascular endothelium and enhance primary and secondary prevention of CVDs. It is becoming increasingly clear that cardiometabolic diseases exhibit sexual dimorphism, underscoring the importance of gaining a deeper understanding of pathologies. To effectively address this issue, it is essential to identify the underlying mechanisms and develop diagnostic tools tailored to each sex. Equally important is the development of preventative and treatment procedures that are personalized to the distinct requirements of each gender. Beyond the well-established cardioprotective effects of AMPK, including antioxidant, anti-inflammatory, and metabolic benefits, there is evidence of cross-talk between AMPK and sex hormones that play a crucial role in sex-dependent cardiovascular homeostasis regulation and thus can also impact the development of cardiometabolic diseases. While the data regarding the impact of AMPK on cardiometabolic regulation is promising, there is still a limited understanding of its dysregulation during the development of cardiometabolic diseases. It is crucial to place even more emphasis on translating these findings to prevent or treat these diseases effectively. Based on the summarized information, AMPK activation might have essential functions in this setting. Future research has to be provided to determine whether AMPK is involved in the metabolic alterations that may influence cardiometabolic diseases, such as metabolic syndrome, T2DM, hypertension, and coronary artery disease, with an emphasis on sex dimorphisms. In general, AMPK is a promising area of research that could lead to effective treatments for cardiometabolic disorders explicitly depending on the sex-hormone-dependent requirements.

## Figures and Tables

**Figure 1 ijms-24-11986-f001:**
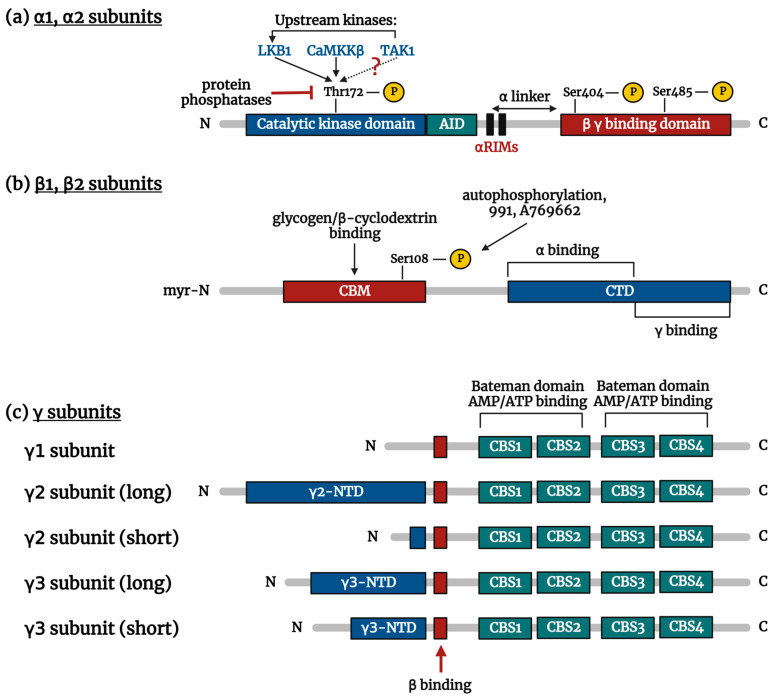
Schematic representation of AMPK subunits: (**a**) α1, α2 subunits; (**b**) β1, β2 subunits; and (**c**) γ subunits—γ1, γ2, and γ3. AID: autoinhibitory domain; CaMKKβ: Ca^2+^-calmodulin-dependent protein kinase kinase beta; CBM: carbohydrate-binding module; CBS: cystathionine β-synthase; CTD: C-terminal domain; LKB1: liver kinase B1; NTD: N-terminal domain; TAK1: transforming growth factor-β-activated kinase 1; αRIMs: α regulatory subunit-interacting motifs.

**Figure 2 ijms-24-11986-f002:**
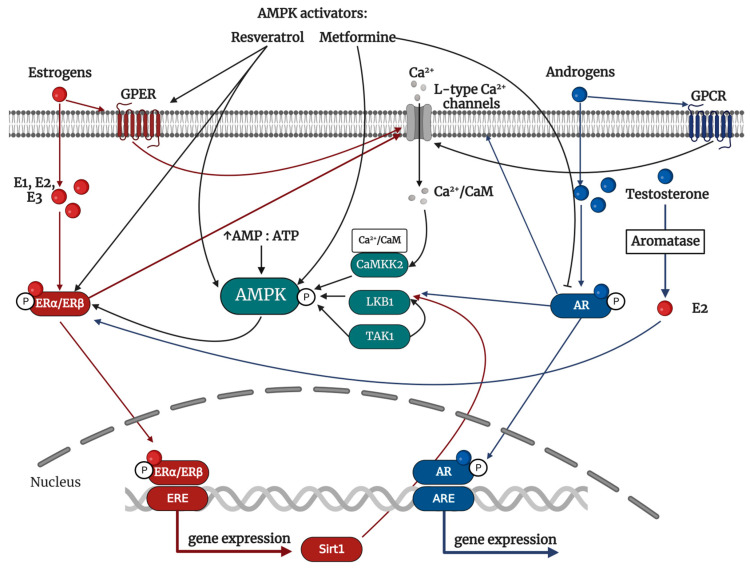
Cross-talk between AMPK and sex hormones. The activation of AMPK occurs when the ratio of AMP:ATP increases and when triggered by upstream kinases, such as CaMKK2, LKB1, and TAK1, in response to various physiological stresses, including hormonal stimuli. Estrogen signaling activates AMPK through Ca^2+^-induced CaMKK2-dependent phosphorylation and Sirt1-dependent LKB1 deacetylation. Additionally, AMPK signaling elevates the expression and activation of ERs. Likewise, androgen signaling activates AMPK through CAMKK2 and LKB1 kinases. AMP: adenosine monophosphate; AMPK: adenosine monophosphate-dependent protein kinase; AR: androgen receptor; ARE: androgen response elements; ATP: adenosine triphosphate; CaM: calmodulin; CaMKK: Ca^2+^-calmodulin-dependent protein kinase kinase; E1: estrone; E2: 17β-estradiol; E3: estriol; ERE: estrogen response elements; ERα: estrogen receptor alpha; ERβ: estrogen receptor beta; GPCR: G protein-coupled receptors; GPER: G-protein-coupled estrogen receptor; LKB1: liver kinase B1; Sirt1: Sirtuin 1; TAK1: transforming growth factor-β-activated kinase 1.

**Figure 3 ijms-24-11986-f003:**
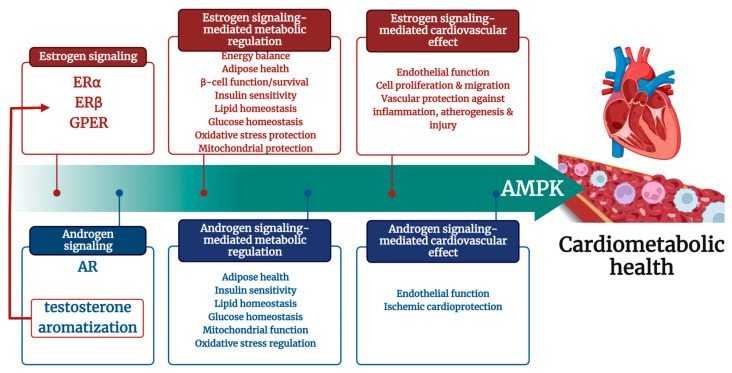
Sex hormones, AMPK cross-talk, and cardiometabolic health. The interaction of AMPK with estrogen and/or androgen signaling can protect the cardiovascular system by enhancing metabolic control and mitochondrial function, decreasing oxidative stress, inflammation, and apoptosis. These improvements ultimately lead to cardiac and endothelial protection. AMPK: adenosine monophosphate-dependent protein kinase; AR: androgen receptor; ERα: estrogen receptor alpha; ERβ: estrogen receptor beta; GPER: orphan—G-protein-coupled estrogen receptor.

## Data Availability

Data sharing is not applicable.

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
