# Peer review of "Sexual Dimorphism in Cardiometabolic Diseases: The Role of AMPK"

_ijms, 2023, doi:10.3390/ijms241511986_

Round 1
Reviewer 1 Report
The manuscript [ID: ijms-2503160] entitled “The Sexual Dimorphism in Cardiometabolic Diseases: The Role of AMPK” presents an overview of the potential role of AMPK in molecular mechanism underlying the sex differences observed in the development and progression of cardiometabolic disease.
The first half of the review is focused upon AMPK and its regulation. The second half of the review involves the specific effects of sex hormones on AMPK activity. The last section of the paper is focused upon the sexual dimorphisms in AMPK regulation of the endothelium. Therefore – the title stating “cardiometabolic diseases” is a little broad compared to the actual focus of the article. Figure 3 presents a more “cardiometabolic” view of the area.
Generally, the review is interesting and focused upon a topic that is scientifically and clinically important – that therefore, should be of interest to readers of this journal.
Generally, the paper is well referenced.
The figures are generally good and informative – and add to the article.
There are some concerns that should be addressed:
Grammar and sentence structure require attention throughout the manuscript. Also, attention to the consistent use of single/plural and past/present tenses.
Specific Issues:
Line54/55 – Reword this sentence – correct type 2 diabetes mellitus
Line 62 – “more presented” should be “more prevalent”
Line 67-69 – “The risk….the misinterpretation that women are more protected than men due to female-specific sex hormone dependent regulation”. In fact, female sex hormones do confer protection – that is why the prevalence of cardiometabolic diseases in is significantly lower in pre-menopausal women – so this is not a misinterpretation. Please re-word/correct this statement.
Line 97 – delete “….,on the other hand”
Line 106 – Change “is bound to” to “can bind to” – as all circulating insulin is not bound.
Line 128 – delete “excellent”
Line 129 – delete “Considering that,”
Line 132 – Clarify what is meant by “IR-worsened glucose”. Please re-word.
In the Section 2.2 – it should be specified that “activation of AMPK is associated with anti-diabetic properties”. This is only stated once.
Line 195 – delete “tissue”
Line 196 – “ In the liver is expressed… “ this sentence has to be re-worded and corrected for grammar.
Line 255 – The sentence “Furthermore, AMP promotes….” Is contradictory. Do the authors really mean to say “Furthermore, elevated concentrations of AMP block the inhibition of AMPK…” ???
Line 432 – delete “Next, the” –
Line 457 – delete “On the other side,”
These are just a few examples of issues with sentence structure and grammar – there are many many more.
see above
Author Response
The manuscript [ID: ijms-2503160] entitled “The Sexual Dimorphism in Cardiometabolic Diseases: The Role of AMPK” presents an overview of the potential role of AMPK in molecular mechanism underlying the sex differences observed in the development and progression of cardiometabolic disease.
- The first half of the review is focused upon AMPK and its regulation. The second half of the review involves the specific effects of sex hormones on AMPK activity. The last section of the paper is focused upon the sexual dimorphisms in AMPK regulation of the endothelium. Therefore – the title stating “cardiometabolic diseases” is a little broad compared to the actual focus of the article. Figure 3 presents a more “cardiometabolic” view of the area.
Answer: Thank you very much for the review; we appreciate your concern about the title. We are aware that the last part of the manuscript is primarily focused on endothelial function. But given the critical role of endothelial function and the fact that endothelial dysfunction is a common significant feature of the development and progression of cardiometabolic diseases, we decided to discuss this topic in terms of AMPK- and sex-hormone-dependent regulation. We know that cardiometabolic diseases are not only about endothelial dysfunction, and it would be appropriate to discuss other aspects, which could even improve the manuscript, but in view of the length of the manuscript, maintaining the interest of the reader and the complexity of the topic, we decided to discuss the endothelial function in more detail in the current manuscript. But based on your review, we also decided to add a chapter discussing cardiac function to the manuscript. In Chapter 6, we added more details regarding the cardiomyocytes (Sexual dimorphism in AMPK regulation of cardiac function).
- Generally, the review is interesting and focused upon a topic that is scientifically and clinically important – that therefore, should be of interest to readers of this journal.
Answer: Thank you very much for your appreciation.
- Generally, the paper is well referenced.
Answer: Thank you very much for your appreciation.
- The figures are generally good and informative – and add to the article.
Answer: Thank you very much for your appreciation.
- There are some concerns that should be addressed:
Answer: Thank you very much for pointing it out. First, we apologize for the mistakes and made corrections as you suggested.
- Grammar and sentence structure require attention throughout the manuscript. Also, attention to the consistent use of single/plural and past/present tenses.
Answer: We apologize for the mistakes and made corrections.
- Specific Issues:
Answer: We apologize for the mistakes and made corrections as you suggested.
- Line54/55 – Reword this sentence – correct type 2 diabetes mellitus
- Line 62 – “more presented” should be “more prevalent”
- Line 67-69 – “The risk….the misinterpretation that women are more protected than men due to female-specific sex hormone dependent regulation”. In fact, female sex hormones do confer protection – that is why the prevalence of cardiometabolic diseases in is significantly lower in pre-menopausal women – so this is not a misinterpretation. Please re-word/correct this statement. → “The risk of CVDs in females is often underestimated, especially in pre-menopausal and menopausal women, due to the misinterpretation that women are still protected due to female-specific sex hormone-dependent regulation.”
- Line 97 – delete “….,on the other hand”
- Line 106 – Change “is bound to” to “can bind to” – as all circulating insulin is not bound.
- Line 128 – delete “excellent”
- Line 129 – delete “Considering that,”
- Line 132 – Clarify what is meant by “IR-worsened glucose”. Please re-word. → “Similarly, ERα deficiency, whether male or female, experiences a decline in glucose regulation, resulting in higher fasting glycemia, impaired glucose tolerance, and reduced insulin-stimulated glucose uptake. Additionally, their lipid metabolism is affected, leading to hyperinsulinemia.”
- In the Section 2.2 – it should be specified that “activation of AMPK is associated with anti-diabetic properties”. This is only stated once. → Thank you very much for pointing it out; we emphasized this fact in the manuscript:
- Abstract: “It seems that AMP-dependent protein kinase (AMPK) may be such a factor since it has the protective role of AMPK in the cardiovascular system, has anti-diabetic properties, and is regulated by sex hormones.“
- Introduction: “In addition, AMPK plays a crucial role in regulating numerous metabolic pathways that are often disrupted in the presence of diabetes mellitus. This includes triggering glucose transport in skeletal muscle and suppressing gluconeogenesis in the liver.“
- Line 195 – delete “tissue”
- Line 196 – “ In the liver is expressed… “ this sentence has to be re-worded and corrected for grammar. → “The α2 isoform is expressed in the liver [44], while the α2, β2, γ2, and γ3 isoforms are specific to skeletal and cardiac muscle [45].“
- Line 255 – The sentence “Furthermore, AMP promotes….” Is contradictory. Do the authors really mean to say “Furthermore, elevated concentrations of AMP block the inhibition of AMPK…” ??? → “Furthermore, AMP promotes the inhibition of phosphatases dephosphorylating AMPK.“
- Line 432 – delete “Next, the” –
- Line 457 – delete “On the other side,”
These are just a few examples of issues with sentence structure and grammar – there are many many more.
Answer: We apologize for the mistakes and made corrections.
Reviewer 2 Report
The versatility of AMPK and its involvement in cardiac disease has been extensively studied. This review is significant because it highlights the sex differences in AMPK involvement, a topic that has only been covered in a limited number of articles. The content is substantial, informative, and intriguing, and the items are well-organized theoretically.
There are no significant concerns but one suggestion. The authors focus their discussion on epithelial cells, with little information or discussion on cardiomyocytes. Since the role of AMPK in cardiomyocytes cannot be ignored and should be well studied in the context of metabolic abnormalities in the heart, there should be a paragraph devoted to discussing the role of AMPK and the importance of sex hormones in cardiomyocytes.
Minor comments:
IR in line 113.
Due to the context of the previous paragraph mentioning the insulin receptor, it might be a little confusing with IR. I suggest spelling out the “IR” again for clarity to the reader.
Line 611-613
“it is well-recognized that endothelial dysfunction is a crucial player in the CVD pathogenesis”
“Maintaining vascular homeostasis is crucial for cardiovascular health”
“its disturbance is a significant risk factor for CVD-related morbidity and mortality”
Please provide references to support these statements.
Author Response
The versatility of AMPK and its involvement in cardiac disease has been extensively studied. This review is significant because it highlights the sex differences in AMPK involvement, a topic that has only been covered in a limited number of articles. The content is substantial, informative, and intriguing, and the items are well-organized theoretically.
There are no significant concerns but one suggestion. The authors focus their discussion on epithelial cells, with little information or discussion on cardiomyocytes. Since the role of AMPK in cardiomyocytes cannot be ignored and should be well studied in the context of metabolic abnormalities in the heart, there should be a paragraph devoted to discussing the role of AMPK and the importance of sex hormones in cardiomyocytes.
Answer: Thank you very much for the review; we appreciate your suggestion. We added a chapter discussing this aspect to the review - 6. Sexual dimorphism in AMPK regulation of cardiac function.
Minor comments:
- IR in line 113. Due to the context of the previous paragraph mentioning the insulin receptor, it might be a little confusing with IR. I suggest spelling out the “IR” again for clarity to the reader.
Answer: We apologize and made corrections as you suggested.
Line 611-613
“it is well-recognized that endothelial dysfunction is a crucial player in the CVD pathogenesis”
“Maintaining vascular homeostasis is crucial for cardiovascular health”
“its disturbance is a significant risk factor for CVD-related morbidity and mortality”
Please provide references to support these statements.
Answer: We apologize and added references to the manuscript as suggested.